# MR and CT Imaging of the Normal Eyelid and its Application in Eyelid Tumors

**DOI:** 10.3390/cancers12030658

**Published:** 2020-03-12

**Authors:** Teresa A. Ferreira, Carolina F. Pinheiro, Paulo Saraiva, Myriam G. Jaarsma-Coes, Sjoerd G. Van Duinen, Stijn W. Genders, Marina Marinkovic, Jan-Willem M. Beenakker

**Affiliations:** 1Department of Radiology, Leiden University Medical Centre, Albinusdreef 2, 2333 ZA Leiden, The Netherlands; M.G.Jaarsma@lumc.nl (M.G.J.-C.); j.w.m.beenakker@lumc.nl (J.-W.M.B.); 2Department of Neuroradiology, Centro Hospitalar e Universitario de Lisboa Central, Rua Jose Antonio Serrano, 1150-199 Lisboa, Portugal; Carolinafpinheiro@gmail.com; 3Department of Radiology, Hospital da Luz, Estrada Nacional 10, km 37, 2900-722 Setubal, Portugal; Paulofsaraiva@gmail.com; 4Department of Ophthalmology, Leiden University Medical Centre, Albinusdreef 2, 2333 ZA Leiden, The Netherlands; S.W.Genders@lumc.nl (S.W.G.); M.Marinkovic@lumc.nl (M.M.); 5Department of Pathology, Leiden University Medical Centre, Albinusdreef 2, 2333 ZA Leiden, The Netherlands; S.G.van_Duinen@lumc.nl

**Keywords:** eyelid tumors, T-staging, eyelid anatomy, tarsus, orbital septum, MRI, CT

## Abstract

T-staging of most eyelid malignancies includes the assessment of the integrity of the tarsal plate and orbital septum, which are not clinically accessible. Given the contribution of MRI in the characterization of orbital tumors and establishing their relations to nearby structures, we assessed its value in identifying different eyelid structures in 38 normal eyelids and evaluating tumor extension in three cases of eyelid tumors. As not all patients can receive an MRI, we evaluated those same structures on CT and compared both results. All eyelid structures were identified on MRI and CT, except for the conjunctiva on both techniques and for the tarsal muscles on CT. Histopathology confirmed the MRI findings of orbital septum invasion in one patient, and the MRI findings of intact tarsus and orbital septum in another patient. Histopathology could not confirm or exclude tarsal invasion seen on MRI on two patients. Although imaging the eyelid is challenging, the identification of most eyelid structures is possible with MRI and, to a lesser extent, with CT and can, therefore, have an important contribution to the T-staging of eyelid tumors, which may improve treatment planning and outcome.

## 1. Introduction

The eyelids correspond to the anterior limit of the orbits. They are muscular-membranous structures [1], forming part of the protective system of the eye. The eyelids have complex anatomy, with each eyelid being constituted of three externally visible regions, namely the external skin, the internal palpebral conjunctiva, and the eyelid margin, all well evaluated with a physical examination. Histologically, however, seven structures [1] are identified in both eyelids, of which the deep structures are not amenable to physical evaluation. The most anterior structure of each eyelid is the skin. Behind the skin, there is the first layer of loose connective tissue. The third layer is the orbicularis oculi muscle, composed of skeletal muscle fibers. The fourth layer, lying behind the orbicularis oculi muscle, is a second layer of loose connective tissue. The fifth layer of each eyelid is a fibro-elastic layer, centrally formed by the tarsal plate and peripherally formed by the orbital septum. The tarsal plate is a firm plate composed of dense connective tissue that helps to maintain the eyelid shape but also containing sebaceous glands called the meibomian glands. The superior tarsus is 8–12 mm in height and attaches to the superior tarsal muscle. The inferior tarsus is smaller, only 3–4 mm in height, and attaches to the inferior septum and inferior tarsal muscle. The orbital septum maintains the intraorbital fat in place and is involved in the ocular and palpebral movements [2]. The orbital septum of both superior and inferior eyelids attaches peripherally to the orbital rim bone, where it is continuous with the periosteum [3]. Centrally the orbital septum attaches to the junction of the inferior tarsal muscle to the tarsal plate in the lower eyelid and to the levator palpebrae aponeurosis in the upper eyelid [1]. Inferiorly to its attachment to the superior orbital septum, the levator palpebrae aponeurosis fuses with the anterior aspect of the superior tarsal plate [4]. Posteriorly, on the most cranial part of the levator palpebrae aponeurosis and at its junction with the levator palpebrae muscle [4], lies the “V” shaped superior transverse (Whitnall) ligament. The sixth layer of the eyelids consists of the tarsal muscles, composed of smooth muscle fibers, acting as eyelid retractors. The superior tarsal muscle, also known as the Müller’s muscle, inserts superiorly at the junction of the levator palpebrae aponeurosis and levator palpebrae muscle and attaches inferiorly to the superior margin of the superior tarsal plate [1]. The inferior tarsal muscle inserts superiorly at the junction of the inferior tarsal plate and inferior septum, and inferiorly attaches to the fascia surrounding the inferior rectus muscle [1]. The most posterior layer of the eyelid is the palpebral or tarsal conjunctiva. The conjunctiva will reflect on the eyeball as bulbar conjunctiva, which is not part of the eyelid (Figure 1). 

Eyelid malignancies can arise from any of the eyelid structures, but most are of cutaneous origin [5]. The most common skin eyelid tumor is the basal cell carcinoma (BCC), followed by squamous cell carcinoma (SCC), sebaceous cell carcinoma, Merkel cell carcinoma, and malignant melanoma [5,6,7]. Eyelid malignancies require specific deliberations as the functional and esthetical impact of surgical treatment can be devastating [6]. 

Accurate staging of an eyelid tumor is based on the Tumor, Node and Metastasis (TNM) classification, and it is important, among others, to help the clinician in the planning of treatment [8,9]. The T-staging of eyelid tumors encompasses determination on whether there is an invasion of the eyelid structures such as the tarsal plate and orbital septum [8,9], which are not clinically accessible and, therefore, imaging can be crucial [10] for an accurate evaluation. 

To our knowledge, eyelid anatomy has never been described on CT, and only a few descriptions are available on MRI [2,11,12,13,14,15,16,17,18]. Similarly, we are not aware of any radiological study depicting which anatomic eyelid structures are invaded by a tumor. The purpose of this manuscript is twofold: (1) To identify the normal anatomy of the eyelid both on MRI and CT, especially the tarsal plates and the orbital septa, (2) To apply this knowledge in tumor patients, comparing imaging data with pathology.

## 2. Material and Methods

### 2.1. Patient Population

This single-center retrospective study was carried out according to the Code of Ethics of the World Medical Association (Declaration of Helsinki) for experiments involving humans and in accordance with the recommendations of the local Ethics Committee (CME LUMC, Leiden University Medical Center, Project number P16.186). Three different groups of patients were evaluated in a total of 41 patients. 

In Group 1, normal eyelid anatomy was assessed on MRI by evaluating MR-images of 19 patients with uveal melanoma (UM) (Table 1). Only the side with UM had been imaged. As the eyelids of these patients were not affected by the UM, they provided a representative view of the healthy anatomy. In 68% of the cases, the right eyelids were evaluated. Both the superior and inferior eyelids of the side available were assessed. Fourteen of the subjects were male (74%), and the median age of the group was 63 years (range 23 to 90). 

In Group 2, normal eyelid anatomy was assessed on CT in 19 patients who received CT scans, including the orbits, with at least one healthy orbit (Table 2). Only the eyelids of one side were assessed, either the eyelids of the non-pathologic orbit or in the case of bilateral normal orbits, one was randomly chosen to be evaluated. In 63% of the cases, the right eyelids were evaluated. Both the superior and inferior eyelids of the side evaluated were assessed. Eleven of the subjects were female (58%). The median age was 45 years (range 19 to 79).

Group 3 consisted of three consecutive patients with different eyelid tumors, who were planned to be treated surgically. These patients received an MRI protocol that had been optimized for the evaluation of eyelid tumors. When available, the CT images were included in the evaluation (Table 3). Two of the subjects were female. The median age was 72 years (range 62 to 81). Patient 1 had a squamous cell carcinoma (SCC) of the palpebral and bulbar conjunctiva at the medial aspect of the left inferior eyelid involving the medial canthus. Clinically, there was evidence of orbital invasion due to abnormal eye movements. Patient 2 had a recurrent SCC of the skin at the medial aspect of the right inferior eyelid, involving the medial canthus and with minimal superior eyelid extension. Clinically, there was no evidence of orbital invasion. Patient 3 had residual melanotic melanoma (MM) at the palpebral conjunctiva of the left superior eyelid, discovered 4 weeks after surgical resection of a MM of medial bulbar conjunctiva of the left inferior eyelid. Clinically there was no evidence of orbital invasion.

### 2.2. MRI Protocol

All MRIs were performed at a 3T MRI (wide bore Ingenia 3T, Philips Healthcare, Best, The Netherlands), using the setup we developed to scan the eyes of UM patients as described in reference [19]. A 4.7 cm surface receive coil (Philips Healthcare, Best, The Netherlands) was used, and the head was supported by a radiotherapy support (MaxSupportTM, Medeo, Schöftland, Switzerland). All patients were asked to close their eyes and try to minimize their eye movements. The eyelids of the patients with eyelid tumors were covered with wet gauze to reduce the susceptibility artifacts due to the tissue-air interface. 

In Group 1 patients, for the evaluation of the healthy eyelid anatomy, the 2D multi-slice (MS) sequences, with 2 mm thickness and an in-plane resolution of at least 0.5 × 0.5 mm^2^ were used [19]. These included T1 (TE/TR:8/718 ms) and T2 (TE/TR:90/1331 ms) weighted-images (WI) and a contrast-enhanced T1-WI (8/764 ms) with spectral presaturation with inversion recovery (SPIR) fat signal suppression. For the analysis of the eyelid anatomy, both sagittal and axial sequences, perpendicular to the eyelid axis, need to be acquired. However, since these scans were primarily acquired to assess the UM, only one of the orientations was available.

Based on the dedicated eye MRI protocol we use for UM patients [19], an optimized protocol for eyelid tumors was developed and used for the evaluation of Group 3 patients. Details of the dedicated eyelid MRI protocol were addressed in Table 4. In this protocol, a 3D T2-weighted scan was performed first and was used as an anatomical reference. Secondly, 2D multi-slice (MS) anatomical sequences were acquired in the sagittal and axial planes, both perpendicular to the main eyelid axis at the level of the tumor. These 2D MS scans consisted of T1-weighted and T2-weighted images, both with and without SPIR fat signal suppression and contrast-enhanced T1-weighted images with SPIR fat signal suppression. Finally, functional sequences were acquired, including diffusion-weighted imaging (DWI) and a dynamic contrast enhanced (DCE) scan with fat signal suppression. The DWI was performed in the sagittal and axial planes, both perpendicular to the main axis of the eyelid at the level of the tumor, while the DCE was acquired in the axial plane but not necessarily perpendicular to the main axis of the eyelid at the level of the tumor. The susceptibility artifacts at the outer eyelid interface were not completely removed by the wet gauze. Therefore, localized volumetric shimming was applied for the patients of Group 3. 

### 2.3. CT Protocol

All CT scans in all Group 2 patients and in patient #1 from Group 3 were performed at a Toshiba Aquilion ONE 320. Volumetric acquisitions of the face, orbits, or paranasal sinuses were acquired, therefore, all including the orbits, without contrast, using the following parameters: a scan range from 40 to 160 mm, 1 rotation per 0.5 s, a reconstruction field of view (FOV) ranging from 171 to 227 mm but mostly of 220 mm, a FC02, FC07, or FC08 filter and a FC30 filter, 120 kV tube voltage, a tube current ranging from 140 to 244 mA but mostly of 200 mA, and a CTDI volume from 12.4 to 15.9 mGy measured on a 16 cm phantom. The normal eyelid anatomy in Group 2 was evaluated using 1 mm soft tissue reconstructions both on the axial and sagittal planes perpendicular to the main axis of the eyelid. On patient #1 from Group 3, the evaluation of the bones was performed using 0.5 mm bone reconstructions both on the axial and coronal planes. 

### 2.4. Image Analysis

Images were evaluated by a Neuro and Head and Neck Radiologist with more than 20 years of experience and by a Neuroradiologist-in-Training with 4 years of experience. 

In Group 1, normal eyelid anatomy on MRI was assessed in a total of 38 eyelids (19 superior eyelids and 19 inferior eyelids). This was evaluated on the axial plane in 53% of the cases and on the sagittal plane in 47% of the cases. In Group 2, normal eyelid anatomy was evaluated on CT in a total of 38 eyelids (19 superior eyelids and 19 inferior eyelids), both in the sagittal and axial planes. These MRI and CT images were compared with histologic slices of normal eyelids. Particular attention was paid to the identification of the superior and inferior tarsal plates and superior and inferior orbital septa, for which a score was created: 1: Not identified; 2: Ill-defined; 3: Well-defined; NA: Non-applicable—in cases where the structure was not included in the available slices (Table 1; Table 2). 

In the 3 patients with different eyelid tumors, Group 3, the tumor localization, and invaded adjacent structures, with special attention for the tarsal plate and orbital septum, were assessed on both sagittal and axial planes and compared with the histopathologic examination after surgery (Table 3). Additionally, in one patient, the standard orbit protocol was compared with the dedicated eyelid protocol. 

The final decision regarding the evaluation in all 3 groups was achieved by consensus.

## 3. Results

In Groups 1 and 2, we evaluated normal eyelid anatomy on MRI and CT, respectively. On MRI, this was best achieved on the T1 and T2 sequences without fat suppression and without contrast. All eyelid layers could be identified, except for the tarsal muscles on CT and for the conjunctiva both on CT and MRI. The skin was isointense or slightly hyperintense to muscle on T1-WI and T2-WI and isodense on CT. Behind the skin, the layer with loose connective tissue had fat signal intensity and density on MRI and CT, respectively, not always being visualized on CT. The orbicularis oculi muscle was well recognized on MRI [17] and CT and seen extending peripherally beyond the edge of the anterior orbital rim. The second layer of loose connective tissue lying behind the orbicularis oculi muscle had the same imaging characteristics as the first connective tissue layer. The fifth layer of each eyelid was formed centrally by the tarsal plate and peripherally by the orbital septum. The superior and inferior tarsal plates appeared as a posterior concave or crescent-shaped line with several dots with signal intensity [15] and density of fat on MRI and CT, respectively, due to the sebaceous content of the meibomian glands. Although this characteristic pattern with several dots was more readily visible on MRI, it can be recognized on CT. On MRI, the superior and inferior orbital septa appeared as hypointense on T1-WI and hypointense on T2-WI, contrasting with the surrounding hyperintense fat [2,13]. On CT, they were hyperdense, in opposition to the adjacent hypodense fat. The sixth layer of the eyelids consists of the tarsal muscles, identified on MRI [2,17] and not identified on CT. The most posterior layer of the eyelid, the conjunctiva, was not seen either on MRI or on CT (Figure 2, Figure 3 and Figure 4).

In Group 1, the visibility of the superior and inferior tarsal plates and orbital septa was scored on MRI (Table 1). Both the superior and inferior tarsal plates were identifiable in 94% of the subjects, being the superior tarsal plate well-defined in 78% and the inferior tarsal plate well-defined in 67% of the subjects. The superior tarsus was easier to identify on the axial plane. The inferior tarsus was equally well visible on the axial and sagittal planes (Figure 2A,E,F, and Figure 3A–D). The superior septum was always visible, being well-defined in 92% of the subjects. The inferior septum was visible in 91% of the subjects, but it was well-defined in only 36% of the subjects. The superior and inferior septa were easier to identify on the sagittal plane (Figure 2A,B,E,F, and Figure 4A–D). Orbital septa and tarsal plates were more difficult to identify when the slices were not acquired perpendicular to the main axis of the eyelid, and when movement artifacts were present. In Group 2, and similarly to Group 1, the superior and inferior tarsal plates and orbital septa were scored (Table 2). The superior tarsus was always visible, being well-defined in 63% of the subjects. The inferior tarsus was visible in 84% of the subjects and was well-defined in 53% of the subjects. Both the superior and inferior tarsal plates were better depicted on the axial plane than on the sagittal plane (Figure 2G and Figure 3E,F). The superior septum was visible in 89% of the subjects and well-defined in 47% of the subjects. The inferior septum was visible in 68% of the subjects, but well-defined only in 11% of the subjects. The superior and inferior septa were easier to identify on the axial plane (Figure 2C,G and Figure 4E,F) than on the sagittal plane. 

In Group 3, tumor extension of 3 patients with different eyelid tumors was determined through image analysis, both on MRI and CT. The MRI of patient #1 was performed with the dedicated eyelid protocol. It showed a heterogeneous enhancing lesion of the medial aspect of the inferior eyelid on the left (Figure 5). The inferior tarsal plate (Figure 5A,B,D,F) and inferior septum (Figure 5A,B,F) were invaded, and so was the medial palpebral ligament region (Figure 5D,E). The tumor grew posteriorly, invading the orbit and reaching the region of the insertion of the inferior rectus muscle at the globe (Figure 5A–C). Due to the location of the tumor, adjacent to the medial orbital bony wall, a CT scan was also performed, but no bone invasion was noticed either on CT or on MRI. CT was able to demonstrate septal and orbital invasion as well, but underperformed compared to MRI and could not depict tarsal invasion. Due to the presence of orbital invasion, both clinically and radiologically, an eyelid-skin sparing orbital exenteration was performed. The final histopathological examination revealed a well-differentiated squamous cell carcinoma at the epithelium of the palpebral conjunctiva, growing anteriorly invading the septum and posteriorly into the intraorbital fat, surrounded by a diffuse inflammatory infiltrate. No perineural or angioinvasive extension was seen (Figure 5H,I). 

In patient #2, initially, a MRI with a standard orbit protocol was performed, followed four days later by a MRI with a dedicated eyelid protocol (Figure 6). Both showed an enhancing lesion at the medial aspect of the inferior right eyelid. With the dedicated MRI protocol invasion of the inferior tarsal plate and medial palpebral ligament region was suspected (Figure 6A–C). The relation of the tumor with the inferior tarsal plate was much more difficult to assess with the standard orbit protocol (Figure 6D–F). The medial wall of the orbit was intact on MRI. Tumor excision was performed with direct defect closure. The final histopathological examination revealed a good/moderately differentiated squamous cell carcinoma of the skin of the eyelid, with free surgical excision margins. No perineural or angioinvasive extension was found (Figure 6G,H).

In patient #3, the MRI with a dedicated eyelid protocol (Figure 7) showed post-surgical changes at the medial inferior left eyelid (not shown), due to resection of a melanotic melanoma of the bulbar conjunctiva of the medial inferior eyelid, performed 4 weeks earlier. MRI failed to show the residual/recurrent tumor at the palpebral conjunctiva of the superior left eyelid, depicting intact tarsal plate and orbital septum (Figure 7A–C). The second surgery included removal of the total palpebral conjunctiva, tarsal plate, and margin of the upper eyelid, preserving the upper eyelid skin and part of the orbicularis muscle. Histopathology showed epithelioid cell melanoma confined to the conjunctival epithelium, with free margins (Figure 7D–F). Further surgery was then performed one week later with eyelid reconstruction using a free tarsoconjunctival graft from the contralateral eyelid. 

## 4. Discussion

Staging of an eyelid tumor is based on the TNM classification and is a critical element in determining the appropriate treatment, a key factor defining prognosis, and will assist in the evaluation of the results of the treatment [8,9,21,22]. Regarding eyelid tumors, different T-stagings are applied depending on the type of tumor and on the layer of origin of the tumor within the eyelid. Eyelid carcinomas, including the basal cell carcinoma, squamous cell carcinoma, sebaceous carcinoma, and other rare carcinomas such as all varieties of sweat gland carcinoma, have a T-staging. Eyelid melanomas, depending on whether they arise within the skin or the conjunctiva, are staged according to the classification for skin melanoma or conjunctival melanoma, respectively. Merkel cell carcinoma is staged using the Merkel cell carcinoma staging system. These different T-stagings encompass determination of tumor dimensions and evaluation of nearby structures’ invasion, namely the tarsal plate, orbital septum, orbit, globe, lacrimal sac/nasolacrimal duct, orbital walls, paranasal sinuses, and the brain [8,9]. Both dimensions and evaluation of nearby structures’ invasion are not always possible through physical examination alone [6,23], with imaging needed for an accurate assessment [9]. On the one hand, according to the American Joint Committee on Cancer (AJCC) Cancer Staging Manual Eighth Edition [9], T-staging of eyelid tumors is assessed through clinical evaluation and/or after biopsy, but with imaging also playing a role in specific situations to assess invasion of the orbit, of the periorbital structures, and perineural spread. On the other hand, according to the Union for International Cancer Control (UICC) TNM Classification of Malignant Tumours Eighth Edition [8], T-staging of eyelid malignancies is assessed only by physical examination or after excision, with no mention of imaging. Although the AJCC considers imaging in some situations regarding the T-staging of eyelid carcinomas and conjunctival melanomas, imaging evaluation of the tarsal plates and orbital septa is never specifically mentioned [8,9]. 

While imaging characterization of orbital tumors often includes mention of the integrity of the globe, orbital bone walls, paranasal sinuses, and the brain [24,25,26,27], one seldom finds references to the eyelid [2,17,18]. To evaluate tumoral invasion of the eyelid structures, in particular of the tarsal plate and orbital septum, which have important therapeutic implications [28,29], accurate knowledge of the complex eyelid anatomy is, therefore, required.

Normal eyelid anatomy has scarcely been a subject of published material, both in MRI [2,11,12,13,14,15,16,17] and CT. To our knowledge, there has been no publication addressing this subject on CT. For the correct analysis of MR and CT data, sagittal and axial images should be obtained, perpendicular to the main eyelid axes. A standard orbit MRI protocol, acquired with a head coil, has a suboptimal resolution, as shown in patient #2 of Group 3, and, therefore, a dedicated eyelid MRI protocol should be employed. One of the main elements of such a protocol is the use of a surface coil, which allows for high-resolution imaging, with an in-plane resolution of <0.5 × 0.5 mm^2^. On CT, a slice thickness reconstruction of 1 mm is suitable for the evaluation of the eyelids. 

Our evaluation showed that all eyelid layers could be identified, except for the tarsal muscles on CT and for the conjunctiva both on MRI and CT. 

Evaluation of invasion of the tarsal plate and the orbital septum is part of the T-staging of most eyelid tumors, and their invasion has direct therapeutic implications. On the one hand, in a tumor confined to the eyelid, which is treated with local resection and reconstructive surgery [6,28,30], knowledge about the presence of tarsal invasion preoperatively is indispensable in planning surgical reconstruction, and adequate information cannot be obtained solely by physical examination. On the other hand, when the orbital invasion is present, an orbital exenteration must be considered [27,29,31]. Notice that in tumors arising from an eyelid layer in front of the orbital septum, such as tumors of the skin of the eyelid, orbital invasion occurs via invasion of the orbital septum, while tumors of the conjunctiva will have direct access to the intraorbital contents since the conjunctiva is located behind the septum limit. Although orbital invasion can be suspected clinically, for example, when signs of eye muscle involvement, such as strabismus or diplopia [30,32], are present, image-based evaluation is necessary, especially in case of non-clinical suspected orbital invasion [25,32]. The recognition of the tarsal plates and orbital septa, both on MRI and CT, is, therefore, crucial in order to evaluate whether they are invaded by an eyelid tumor and, therefore, their visualization on MRI and CT was scored. Although MRI has a higher soft-tissue resolution, the results using CT were positively surprising. 

Regarding the tarsal plates, our results show that they are visible most of the time on both CT and MRI, although with a better definition on MRI. The superior tarsal plate was visible in 94% of the subjects on MRI and was always visible on CT. This was probably due to the fact that the patients of Group 1 were not scanned specifically to assess the eyelids, resulting in suboptimal MR-images for their evaluation, with only either axial or sagittal images available and not always perpendicular to the main axes of the eyelid. On the contrary, on CT, both axial and sagittal reformats perpendicular to the main axes of the eyelid were available. The inferior tarsal plate was always easier to depict on MRI, even when suboptimal MR-images were used. Axial planes should be chosen over sagittal planes to identify the tarsal plates, with the superior tarsal plate being more readily recognizable due to its larger size. The superior and inferior tarsal plates continue laterally to the orbital bone rim as medial and lateral palpebral ligaments, both possible to identify on MRI [13,14] (Figure 3A,B). 

Regarding the orbital septa, our results show both to be mostly visible on MRI and CT, but a better definition was generally achieved on MRI. The superior and inferior orbital septa were easier to identify on the sagittal plane on MRI, while on CT, their identification was easier on the axial plane. Both on MRI and CT, the inferior septum is more difficult to see than the superior septum. This is not only due to its smaller size, but also because the shape of the inferior orbital septum changes with aging [33,34], with the protrusion of the intraorbital fat anteriorly displacing the inferior septum against the orbicularis oculi muscle and making it difficult to tell them apart (Figure 2E–G). 

The literature on eyelid tumors mentions the role of imaging, either MRI or CT, mainly in large tumors in which orbital invasion is present [25,30]. With this work, we were able to demonstrate the potential use of MRI, and to a lesser extent of CT, in the identification of tumor invasion of small anatomical structures in the eyelid, important in the TNM classification, such as the tarsus and the orbital septum, which are not clinically accessible. Because of its superior soft-tissue contrast and spatial resolution, MRI is the modality of choice to evaluate the extension of an eyelid tumor. Moreover, with MRI, diffusion, and perfusion-weighted imaging will help to differentiate between the malignant tumor and potentially surrounding inflammation. The use of a surface coil is optimal for the evaluation of tumor extension in the eyelid, but also for the invasion of adjacent structures by an eyelid tumor. Although surface coils are less suitable for assessing the deeper aspect of the orbit, such as the orbital apex, imaging thereof is generally not necessary in the context of an eyelid tumor. The perineural spread is the exception, and, ideally, an axial contrast-enhanced T1-weighted with fat signal suppression using a head coil should be performed in order to image the orbital apex and cavernous sinus adequately. Additionally, because part of the contralateral eyelid is sometimes used to reconstruct the eyelid where the tumor had been resected [7], an additional axial T1-WI from the contralateral normal eyelid could also be acquired to aid with the surgery planning. These images could furthermore be used as a reference to better interpret the pathologic side. CT should mainly be used as a complementary technique in the evaluation of bone invasion [24,27,34]. In cases where MRI cannot be performed, CT has some potential in the evaluation of tumor extension, as it allows for the visualization of most of the eyelid structures, although further studies are needed to fully establish the clinical value of CT for eyelid tumors.

In this study, we assessed the usefulness of MRI and CT images in the delineation of eyelid tumors and their relation to the surrounding eyelid structures, by confronting image data of three patients with surgical findings and histopathology. The MR-images of patient #1 showed invasion of the inferior tarsal plate, inferior septum, and orbit as far dorsally as the insertion of the inferior rectus muscle at the globe. While septum and intraorbital invasion could also be visualized on CT, the tarsal invasion was not evident. Histopathology confirmed both the septum and orbital invasion, but the tarsus invasion was not accessed and, therefore, could not be confirmed but also not excluded. That is because although with MRI the whole tumor is evaluated, histopathology slices can only be made in one plane, mainly planned to evaluate whether free surgical margins exist, not always matching those of imaging and not covering the whole tumor. As a result, on the evaluation of whether a specific eyelid structure is invaded by the tumor, histopathology can only act as the gold standard when positive, while when histopathology findings are negative, they do not necessarily invalidate MRI findings. Based on the MRI of patient #2, invasion of the inferior tarsal plate was suspected. This could not be confirmed on histopathology, again due to the lack of spatial correlation between imaging and histologic slices. The MRI of patient #3 did not show the tumor, clinically located at the palpebral conjunctiva of the superior eyelid, meaning that the superior tarsal plate and superior septum were intact. Histopathology confirmed that the tumor was superficial and confined to the conjunctival epithelium.

When evaluating CT or MR-images with eyelid tumors, radiologists must be familiar with both the complex eyelid anatomy and its imaging and with criteria in tumor staging. Moreover, the tumor location must be known before the MRI-exam, by seeing either the patient or the patient’s photograph, in order to correctly plan the adequate MRI scans, as these small lesions are often not visible on the low-resolution images that are used to plan the higher resolution acquisitions. As in other areas of medicine, a multidisciplinary approach should be encouraged, as contributions from both radiologists and ophthalmologists may lead to better information gathering needed in treatment planning, with a positive impact on the patient’s outcome.

Our study has some limitations. First, the evaluation of the healthy eyelid anatomy was done retrospectively, thus a dedicated eyelid MRI protocol was not used. As a result, only one single orientation, either sagittal or axial, was available, and these slices were sometimes not planned perpendicular to the main axes of the eyelid. Furthermore, some technical aspects of the dedicated eyelid protocol, such as the localized shimming, have not been applied to the patients of the healthy eyelid group. Secondly, the local extension of eyelid tumors was only evaluated on three patients. Finally, an accurate correlation between tomographic imaging and histopathological examination is not always possible.

## 5. Conclusions

Despite the small size of the various components of eyelid anatomy and although imaging the eyelid is challenging due to susceptibility and motion artifacts, the delineation of most of the eyelid structures is possible with an optimized MRI protocol, and to a lesser extent with CT. MR imaging is, therefore, important for the assessment of tumor invasion of the tarsal plate and orbital septum, having an important contribution to the T-staging of eyelid tumors, which may improve treatment planning and may have a positive impact on both patients’ short time morbidity and longtime outcome. 

## Figures and Tables

**Figure 1 cancers-12-00658-f001:**
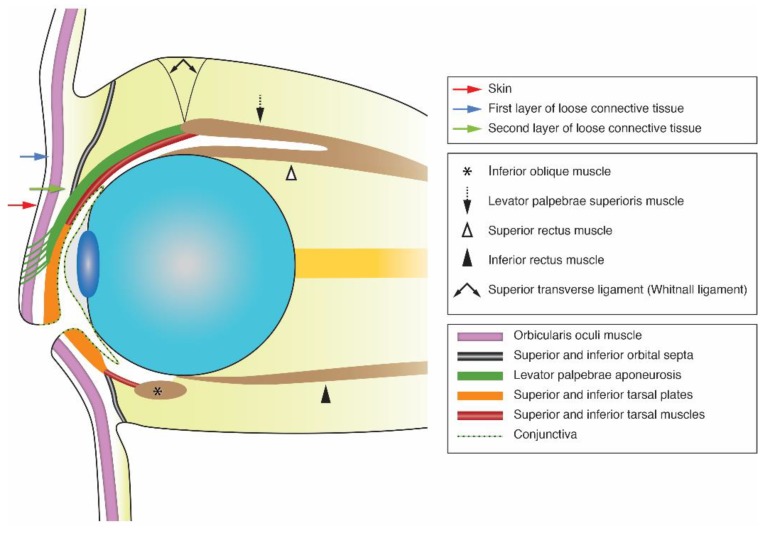
Schematic cross-section of the eyelids and anterior orbital anatomy.

**Figure 2 cancers-12-00658-f002:**
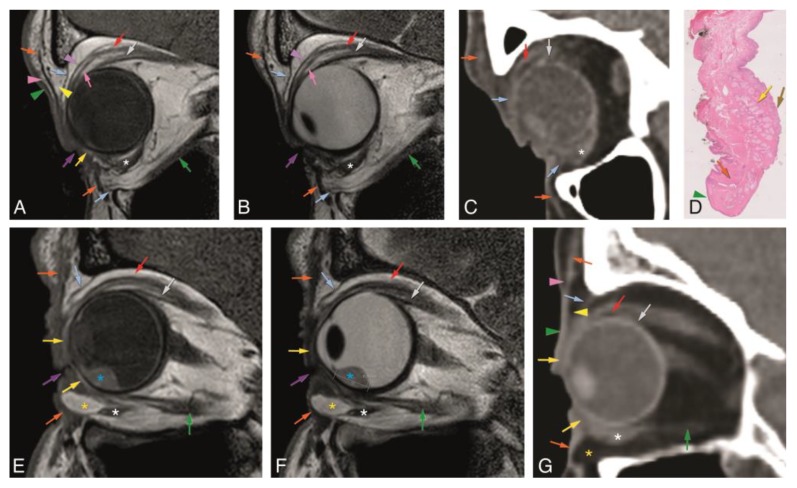
A–D: Normal eyelid anatomy, on the sagittal plane, on MR T1-WI (**A**) and T2-WI (**B**), on CT (**C**), and on histopathological examination (**D**) (1×) obtained from an orbit exenteration with skin preservation. E–G: Normal eyelid anatomy with increasing age changes on the sagittal plane on MR T1-WI (**E**) and T2-WI (**F**), and on CT (**G**). Notice the anterior protrusion of the orbital fat of the inferior eyelid (yellow asterisk), often occurring in older people. Further notice the uveal melanoma (blue asterisk), located inferiorly, at E and F. Purple arrow: palpebral fissure; green arrowhead: skin; pink arrowhead: first layer of loose connective tissue; orange arrow: orbicularis oculi muscle; yellow arrowhead: second layer of loose connective tissue; yellow arrow: superior and inferior tarsal plates; blue arrow: superior and inferior orbital septa; pink arrow: superior tarsal muscle; brown arrow: conjunctiva; purple arrowhead: levator palpebrae aponeurosis; red arrow: levator palpebrae muscle; grey arrow: superior rectus muscle; white asterisk: inferior oblique muscle; green arrow: inferior rectus muscle; yellow asterisk: inferior eyelid fat protrusion; blue asterisk: uveal melanoma.

**Figure 3 cancers-12-00658-f003:**
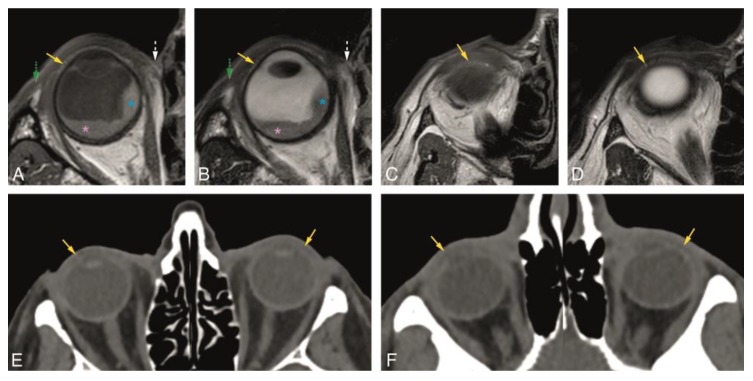
A, B, and E: Normal superior tarsal plate (yellow arrow) on the axial plane on MR T1-WI (**A**), T2-WI (**B**), and on CT (**E**). Notice, at A and B, the uveal melanoma (blue asterisk) with associated retinal detachment (pink asterisk), impossible to differentiate on non-contrast enhanced sequences. C, D, and F: Normal inferior tarsal plate (yellow arrow) on the axial plane on MR T1-WI (**C**), T2-WI (**D**), and on CT (**F**). Yellow arrow: superior and inferior tarsal plates; green dashed arrow: lateral palpebral ligament region; white dashed arrow: medial palpebral ligament region; blue asterisk: uveal melanoma; pink asterisk: retinal detachment.

**Figure 4 cancers-12-00658-f004:**
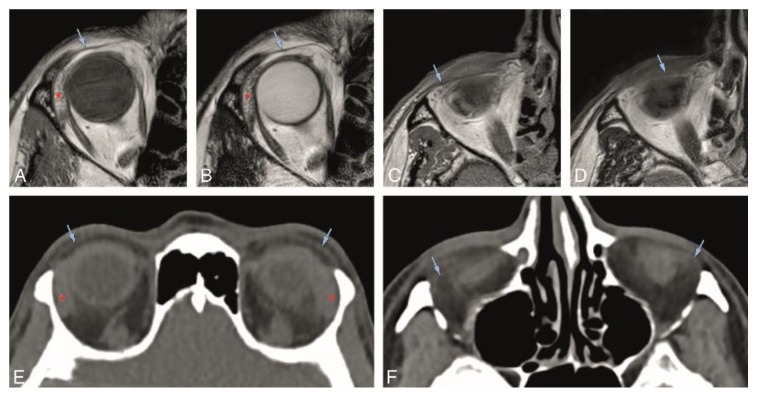
A, B, and E: Normal superior orbital septum (blue arrow) on the axial plane on MR T1-WI (**A**), T2-WI (**B**), and on CT (**E**). C, D, and F: Normal inferior orbital septum (blue arrow) on the axial plane on MR T1-WI (**C**), T2-WI (**D**), and on CT (**F**). Blue arrow: orbital septum; red asterisk: lacrimal gland.

**Figure 5 cancers-12-00658-f005:**
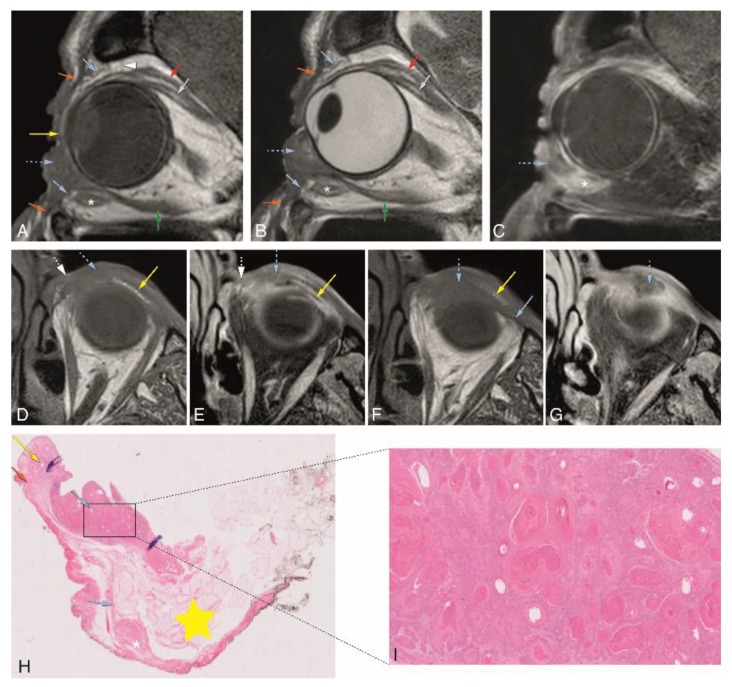
A–I: Patient #1 of Group 3 with a squamous cell carcinoma (SCC) of the left inferior tarsal and bulbar conjunctiva (dedicated eyelid protocol). A–C: Sagittals T1-WI (**A**), T2-WI (**B**), and contrast-enhanced T1-WI with fat signal suppression (**C**) showing the tumor (blue dashed arrow) invading the orbit and reaching the region of insertion of the inferior rectus muscle at the globe. D–G: Axials T1-WI (**D**,**F**) and contrast-enhanced T1-WI with fat signal suppression (**E**,**G**) at the level of the inferior eyelid (slices **D** and **E** are superior to slices **F** and **G**). Notice the tumor (blue dashed arrow) invading the medial inferior tarsal plate (yellow arrow) and medial palpebral ligament region (white dashed arrow) (**D**,**E**), and inferiorly growing behind the septum limit (blue arrow) (**F**,**G**). H–I: Histopathological examination hematoxylin and eosin stain (H&E) (0.5×) (**H**) and histopathological examination H&E stain (5×) (**I**). Notice the tumor (blue dashed arrow) at the epithelium of the palpebral conjunctiva growing anteriorly and invading the septum and posteriorly into the intraorbital fat (**H**). Well-differentiated SCC (**I**). Blue dashed arrow: tumor; orange arrow: orbicularis oculi muscle; yellow arrow: superior and inferior tarsal plates; blue arrow: superior and inferior orbital septa; white dashed arrow: medial palpebral ligament region; yellow star: intraorbital fat; white arrowhead: superior transverse ligament (Whitnall ligament); red arrow: levator palpebrae muscle; grey arrow: superior rectus muscle; white asterisk: inferior oblique muscle; green arrow: inferior rectus muscle.

**Figure 6 cancers-12-00658-f006:**
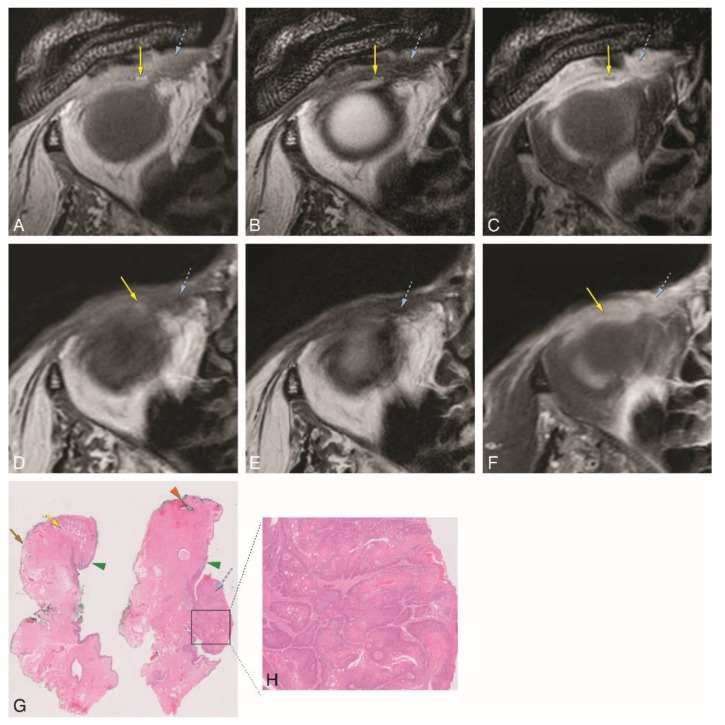
A–H: Patient #2 of Group 3 with a SCC of the skin of the right medial inferior eyelid. Comparison between a dedicated eyelid MRI protocol with a standard orbit MRI protocol and with histopathology. A–C: Dedicated eyelid protocol with axials T1-WI (**A**), T2-WI (**B**) and contrast-enhanced T1-WI with fat signal suppression (**C**) at the level of the inferior eyelid, showing the tumor (blue dashed arrow) involving the medial corner of the eye, with dubious involvement of the inferior tarsal plate (yellow arrow). D–F: Standard orbit protocol with axials T1-WI (**D**), T2-WI (**E**), and contrast-enhanced T1-WI with fat signal suppression (**F**) at the level of the inferior eyelid. Notice the much more difficult identification of the inferior tarsal plate (yellow arrow) and its relation with the tumor (blue dashed arrow). G and H: Histopathological examination H&E stain (0.5×) (**G**) and histopathological examination H&E stain (5×) (**H**). There was no slice available containing simultaneously the tumor and the tarsal plate and, therefore, it was difficult to evaluate tarsal invasion by the tumor. Notice that on (**G**), the left slice includes normal tarsal plate tissue (yellow arrow), but no tumor is seen, while the right slice shows the exophytic tumor (blue dashed arrow) arising from the skin but no tarsal plate tissue is seen. Skin cell proliferation compatible with good/moderately differentiated SCC (**H**). Blue dashed arrow: tumor; green arrowhead: skin; yellow arrow: tarsal plate; brown arrow: conjunctiva; orange arrowhead: eyelid margin.

**Figure 7 cancers-12-00658-f007:**
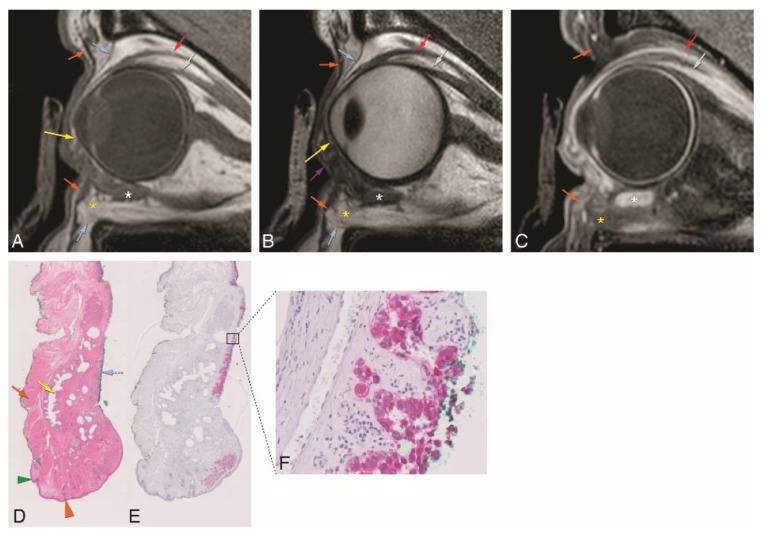
A–F: Patient #3 of Group 3 with a melanotic melanoma of the left superior tarsal conjunctiva (dedicated eyelid protocol). A–C: Sagittals T1-WI (**A**), T2-WI (**B**), and contrast-enhanced T1-WI with fat signal suppression (**C**). It is not possible to visualize the tumor as it is too superficial in the conjunctiva. D-F: Histopathological examination H&E stain (1×) (**D**), histopathological examination Melan-A stain (1×) (**E**), and histopathological examination Melan-A stain (40×) (**F**). Notice the epithelioid cell melanoma (blue dashed arrow) confined to the conjunctiva epithelium. Blue dashed arrow—tumor; purple arrow: palpebral fissure; green arrowhead: skin; orange arrow: orbicularis oculi muscle; yellow arrow: superior and inferior tarsal plates; blue arrow: superior and inferior orbital septa; orange arrowhead: eyelid margin; red arrow: levator palpebrae muscle; grey arrow: superior rectus muscle; white asterisk: inferior oblique muscle; yellow asterisk: inferior eyelid fat protrusion.

**Table 1 cancers-12-00658-t001:** Normal eyelid MRI anatomy Group (Group 1) patients’ data regarding studied eyelid, slice plane evaluated and visualization score of the superior and inferior tarsal plates and of the superior and inferior orbital septa (1: Not identified; 2: Ill-defined; 3: Well-defined; NP: Not possible to evaluate—when the structures to be evaluated were not totally included in the available slices).

Patient	Studied Eyelid	Slice Plane	Superior Tarsus	Inferior Tarsus	Superior Septum	Inferior Septum
1	OS	Axial	3	2	2	1
2	OD	Axial	2	3	NP	NP
3	OD	Sagittal	1	1	3	2
4	OD	Sagittal	3	3	3	3
5	OD	Sagittal	2	2	3	3
6	OD	Axial	3	3	3	NP
7	OD	Sagittal	3	3	3	2
8	OS	Sagittal	3	3	3	3
9	OD	Sagittal	2	3	3	3
10	OS	Sagittal	3	3	3	2
11	OS	Axial	3	NP	3	NP
12	OD	Axial	3	3	NP	NP
13	OD	Axial	3	3	NP	2
14	OD	Axial	NP	2	NP	NP
15	OD	Axial	3	3	3	NP
16	OS	Axial	3	3	NP	NP
17	OD	Sagittal	3	2	3	2
18	OS	Axial	3	2	NP	NP
19	OD	Sagittal	3	3	3	2

OS: Oculus sinister; OD: Oculus dexter.

**Table 2 cancers-12-00658-t002:** Normal eyelid CT anatomy Group (Group 2) patients’ data regarding studied eyelid and visualization score of the superior and inferior tarsal plates and of the superior and inferior orbital septa both on the sagittal and axial planes (1: Not identified; 2: Ill-defined; 3: Well-defined; NP: Not possible to evaluate—when the structures to be evaluated were not totally included in the available slices).

Patient	Studied Eyelid	Superior Tarsus		Inferior Tarsus		Superior Septum		Inferior Septum	
		Sag	Ax	Sag	Ax	Sag	Ax	Sag	Ax
1	OD	2	3	2	2	2	2	2	2
2	OD	3	3	2	2	2	3	1	1
3	OD	2	2	2	2	3	3	2	2
4	OS	2	3	2	3	2	3	2	3
5	OD	3	3	3	3	3	3	1	1
6	OS	1	2	1	1	2	2	1	1
7	OS	1	2	2	2	3	3	2	3
8	OD	3	2	1	1	2	2	1	2
9	OS	1	2	1	1	3	3	2	2
10	OD	2	3	2	2	1	1	1	1
11	OD	2	3	2	3	2	3	2	2
12	OS	2	2	2	3	2	2	2	2
13	OS	3	3	3	3	1	2	1	1
14	OD	1	2	1	2	2	2	1	2
15	OD	2	3	2	3	2	1	1	1
16	OD	2	3	2	3	2	2	1	2
17	OD	2	3	2	3	2	3	2	2
18	OD	2	2	3	3	3	3	2	2
19	OS	3	3	2	3	1	1	2	2

OS: Oculus sinister; OD: Oculus dexter; Sag: Sagittal; Ax: Axial.

**Table 3 cancers-12-00658-t003:** Group 3 patients’ data regarding pathologic eyelid, clinical tumor localization, final histopathology, presence of septum invasion, and presence of tarsal invasion.

Patient	Pathologic Eyelid	Tumor localization (Clinical)	Histopathology	Septum Invasion (Imaging)	Tarsal Invasion (Imaging)
1	OS	Medial inferior eyelid palpebral and bulbar conjunctiva	SCC	Yes	Yes
2	OD	Medial inferior and superior eyelid skin	SCC	No	Suspected
3	OS	Superior eyelid palpebral conjunctiva	MM	No (tumor not seen)	No (tumor not seen)

OS: Oculus sinister; OD: Oculus dexter; SCC: Squamous cell carcinoma; MM: Melanotic melanoma.

**Table 4 cancers-12-00658-t004:** Scan parameters of the sequences of the dedicated eyelid MRI protocol. Both the multi-slice (MS) and the diffusion-weighted imaging (DWI) sequences are planned perpendicular to the eyelid at the location of the tumor, while the dynamic contrast enhanced (DCE) scan is acquired not necessarily perpendicular to the eyelid at the location of the tumor.

Scan Name	Voxel Size (mm^3^)	FOV (mm^3^)	Echo Train Length	TE(ms)/TR(ms)/Flip or Ref. Angle (deg)	Fat Supr.	Avg.	Scan Time (mm:ss)	Additional Parameters
3D TSE T2 SPIR	0.8 × 0.8 × 0.8	50 × 81 × 40	117	293/2500/35	SPIR	2	03:35	4 REST slabs to prevent foldover
MS TSE T1	0.5 × 0.5 × 2.0	100 × 100 × 24	6	8/400/180	-	1	0:43	50 mm foldover suppression
MS TSE T2	0.4 × 0.4 × 2.0	100 × 100 × 24	17	90/1131/120	-	2	01:12	
MS TSE T1 SPIR	0.5 × 0.5 × 2.0	100 × 100 × 24	6	8/458/180	SPIR	1	0:50	50 mm foldover suppression
MS TSE T2 SPIR	0.4 × 0.4 × 2.0	100 × 100 × 24	17	90/1249/120	SPIR	2	01:20	
DWI (TSE)	1.25 × 1.4 × 2.4	100 × 100 × 24	single shot	64/5759/50	SPIR	5	03:21	B = 0, 800 s/mm^2^ 24 mm foldover suppression
DCE	1.0 × 1.0 × 2.0	78 × 78 × 40	single shot	2.5/4.8/13	PROSET11	1	4:33	3 s/dynamic using TWIST

TWIST: Time-resolved angiography with interleaved stochastic trajectories [20].

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
