# Peer review of "MR and CT Imaging of the Normal Eyelid and its Application in Eyelid Tumors"

_cancers, 2020, doi:10.3390/cancers12030658_

Round 1
Reviewer 1 Report
The purpose and hypothesis is a little confusing. The authors need to improve the presentation of the results to more clearly follow the logic and purpose of this study.
The authors state that because imaging can evaluate extent of invasion that this will help with T-staging, treatment planning and outcome. I think this is too strong of a conclusion. This is not what the study assessed. The study did not compare the T-stage, treatment plans/revisions, and outcomes of patients who did receive imaging vs those that did not. Malignant eyelid lesions are excised, and the excision will provide accurate T-staging (this is how the validation of the AJCC staging system was performed). Many eyelid tumors provide sufficient information on clinical examination to completely plan treatment. Although I agree, that in a ‘select’ group of patients – ie those in which the septum and tarsal involvement is uncertain on clinical examination that treatment planning “may” be improved. And this “may” improve outcomes.
In the introduction, the authors state the layers of the eyelid. But they do not qualify which eyelid (upper and lower) and where on the eyelid 15mm from margin, 5mm from margin… This needs to be done if eyelid layers are being presented since the layers vary.
The authors state that surgical planning for eyelid lesions is based solely on clinical examination. This is incorrect. If orbital and or bone involvement is a possibility, patients get imaging in almost every centre in a first world country. This paragraph needs to be corrected as it overstates the underuse of imaging.
The authors stated that in group 1, the CT arm, the side evaluated was selected as the one with better imaging quality. This is an inappropriate method which will overestimate the information that can be obtained from a CT scan. The authors need to randomly select a side.
It is not clear if the same methodology of imaging was applied to all patients. If the technique, example ‘slice thickness’ varied than comparison of imaging modalities (MRI vs CT) is not appropriate since there is expected to be variation within a single imaging modality if various methods were used. The authors mentioned this in the discussion as a weakness. But the methods section also needs to be clarified as to who received what imaging.
Using both the upper and lower eyelid is not ideal due to the correlation in imaging findings within patients. This is more of an issue when formal statistical tests are used since statistical assumptions are being violated, but can lead to erroneous conclusions even with descriptive data. I would recommend performing a quick sensitivity analysis in which only 1 eyelid (upper or lower) is evaluated. Which eyelid is selected should be random. And seeing if this leads to any difference in the reported outcomes.
I would caution the authors in repeatedly stating that pathology is not the gold standard in assessing invasion. I believe the authors are slightly overstating the limitations of pathology. If the surgeons feel that tarsal invasion is a critical finding, was this relayed to the pathologist? The radiaologists knew in advance what they were looking for.
The authors need to state that the 3 cases with tumors that were sent in were imaged because of the advanced nature of the lesion and possibly the location and therefore don’t represent a ‘standard’ eyelid tumor. As well it is difficult to assess how medial these lesions were and whether they were exiting the central eyelid anatomy and were entering the medial canthus anatomy which has a completely different histological layers.
Author Response
Response to Reviewer 1 Comments:
We thank the Reviewer for the evaluation of the manuscript entitled “MR and CT Imaging of the Normal Eyelid and its application in Eyelid Tumors” we submitted to Cancers. You will find that the comments and critics have been taken in utmost consideration. At the “Responses” the “Lines” we mention are the Lines in the Revised Version of the Manuscript (CLEAN Version – without trackchanges). We have addressed them in the following way:
Point 1:English language and style are fine/minor spell check required.
Response 1:It was done.
Point 2: Introduction, Research design, Methods, Results and Conclusions must be improved.
Response 2: Introduction, Materials and Methods, Results and Conclusions were improved according to both reviewers’ suggestions.
Point 3:The purpose and hypothesis is a little confusing. The authors need to improve the presentation of the results to more clearly follow the logic and purpose of this study.
Response 3:The purpose of our study is described in the Abstract and in Lines 82-85.
For the staging of most eyelid tumors tarsal and orbital septum invasion need to be evaluated, but that is not possible by physical examination. MRI is the best exam to evaluate tumor extension inside the orbit and in the periorbital structures, and it also seems the best imaging modality to evaluate its extension within the eyelid. In order to do so we need to be able to delineate the eyelid structures on MRI. And that is the first purpose of our study. Because sometimes patients cannot perform MRI, we decided also to evaluate if the normal eyelid structures were possible to delineate on CT. Finally we checked in 3 patients with eyelid tumors if we could assess tarsal and septal invasion by the tumors.
The Results were changed: The description of how all eyelid layers appear on MRI and CT was moved from the Discussion to the Results. See Lines 197-214.
Point 4:The authors state that because imaging can evaluate extent of invasion that this will help with T-staging, treatment planning and outcome. I think this is too strong of a conclusion. This is not what the study assessed. The study did not compare the T-stage, treatment plans/revisions, and outcomes of patients who did receive imaging vs those that did not. Malignant eyelid lesions are excised, and the excision will provide accurate T-staging (this is how the validation of the AJCC staging system was performed). Many eyelid tumors provide sufficient information on clinical examination to completely plan treatment. Although I agree, that in a ‘select’ group of patients – ie those in which the septum and tarsal involvement is uncertain on clinical examination that treatment planning “may” be improved. And this “may” improve outcomes.
Response 4:The authors agree with the Reviewer. We changed the Abstract to: “can therefore have an important contribution on the T-staging of eyelid tumors, which may improve treatment planning and outcome.” We changed the conclusion to: “MR imaging is therefore important for the assessment of tumor invasion of the tarsal plate and orbital septum, having an important contribution on the T-staging of eyelid tumors, which may improve treatment planning and may have a positive impact on both patients’ short time morbidity and longtime outcome.” We also understand that in many eyelid tumors clinical examination provides sufficient information to completely plan treatment at an individual basis. But when looking at a group of patients with eyelid tumors, it is important that they all are staged and in a uniform way, in order to evaluate the outcomes of treatments. Also, to us it seems correct to use image, in particular MRI, as a validation tool that there is no deep invasion, as it currently happens with other Head and Neck tumors such as a tong tumor.
Point 5:In the introduction, the authors state the layers of the eyelid. But they do not qualify which eyelid (upper and lower) and where on the eyelid 15mm from margin, 5mm from margin… This needs to be done if eyelid layers are being presented since the layers vary.
Response 5:This has been clarified in the manuscript as the extensive description of the eyelids anatomy that was on the Discussion was moved to the Introduction.Regarding “where on the eyelid 15 mm from margin 5 mm from margin” we are describing all vertical extension of the eyelid, as we are describing the tarsus, but also the orbital septum, levator palpebrae aponevrosis…
Point 6:The authors state that surgical planning for eyelid lesions is based solely on clinical examination. This is incorrect. If orbital and or bone involvement is a possibility, patients get imaging in almost every centre in a first world country. This paragraph needs to be corrected as it overstates the underuse of imaging.
Response 6:The authors agree with the Reviewer and changed it. See Lines 370-378. The UICC (Union for International Cancer Control)does not mention image for the T-staging of any eyelid tumor, stating it is done by physical examination and/or after biopsy. The AJCC (American Joint Committee on Cancer)does consider Image.
Point 7:The authors stated that in group 1, the CT arm, the side evaluated was selected as the one with better imaging quality. This is an inappropriate method which will overestimate the information that can be obtained from a CT scan. The authors need to randomly select a side.
Response 7:The authors agree with the Reviewer and scored again the CT’s in Group 2. See Lines 106-8: “Only the eyelids of one side were assessed, either the eyelids of the non-pathologic orbit or in case of bilateral normal orbits one was randomly chosen to be evaluated.” Table 2, the Results and Discussion were updated.
Point 8:It is not clear if the same methodology of imaging was applied to all patients. If the technique, example ‘slice thickness’ varied than comparison of imaging modalities (MRI vs CT) is not appropriate since there is expected to be variation within a single imaging modality if various methods were used. The authors mentioned this in the discussion as a weakness. But the methods section also needs to be clarified as to who received what imaging.
Response 8:
Group 1 patients received MRI with a protocol described in Lines 140-146. The MRI’s of all patients within this group had the same imaging parameters such as slice thickness and contrast parameters (and the same used in Group 3). The only thing that varied in the MR protocol of these patients is that for each patient only one single orientation, either sagittal or axial, was available – and that is what we point as a weakness. For the rest the protocol is optimal.
Group 2 patients received CT with a protocol described in Lines 167-176. In line 258: “in all Group 2 patients and in patient #1 from Group 3” was added for clarification as suggested.Within the patients from Group 2 small variations in scanning parameters do exist, since the CT was acquired for a specific clinical question, which for example required an extended field of view.
Although the MRI protocol in Group 1 was suboptimal becausefor each patient only one single orientation was available, the results in terms of identification of the eyelid structures were good, and MRI performed better than CT, meaning that if the optimal protocol would be applied we would expect even better results.
Group 3 patients – with eyelid tumors - received MRI with a protocol described in Lines 147-160 and in Table 4. It was an optimal MRI protocol. All patients had MRI’s with the same parameters within this group.
Point 9:Using both the upper and lower eyelid is not ideal due to the correlation in imaging findings within patients. This is more of an issue when formal statistical tests are used since statistical assumptions are being violated, but can lead to erroneous conclusions even with descriptive data. I would recommend performing a quick sensitivity analysis in which only 1 eyelid (upper or lower) is evaluated. Which eyelid is selected should be random. And seeing if this leads to any difference in the reported outcomes.
Response 9:
The aim of this manuscript is to assess the visibility of the different eyelid structures on MRI and CT, being the exact percentages not the key outcome, but more the general visibility of the different structures.
Furthermore, the subjects of Groups 1 and 2 are a representative selection of the general population (and have no known eyelid pathology), so also therefore no bias towards is expected by including both eyelids.
We confirmed this for the evaluation of the MRI-findings, by randomly splitting the group in 2 and evaluating only the superior data in the first subset and only the inferior data in the second subset (listing well-defined; ill-defined; not visible). No significant differences are found between the full and the sub-set evaluation:
Superior tarsal plate: full set: 78%;16%;6% subset: 70%;20%;10%
Superior septum: full set: 92%;8%;0% subset: 89%;11%;0%
Inferior tarsal plate: full set: 67%; 29%;4% subset: 78%;22%;0%
Inferior septum: full set: 64%;36%;0% subset: 67%;33%;0%
Point 10:I would caution the authors in repeatedly stating that pathology is not the gold standard in assessing invasion. I believe the authors are slightly overstating the limitations of pathology. If the surgeons feel that tarsal invasion is a critical finding, was this relayed to the pathologist? The radiaologists knew in advance what they were looking for.
Response 10:We erased from the text in Line 484: “and therefore pathology cannot always act as the gold standard for the evaluation of the MRI findings”.
Point 11:The authors need to state that the 3 cases with tumors that were sent in were imaged because of the advanced nature of the lesion and possibly the location and therefore don’t represent a ‘standard’ eyelid tumor. As well it is difficult to assess how medial these lesions were and whether they were exiting the central eyelid anatomy and were entering the medial canthus anatomy which has a completely different histological layers.
Response 11:The 3 presented cases were not selected because "of the advance nature of the lesion". When the MRI protocol was finalised, these were the first three eyelid tumor patients which presented at our ophthalmology clinic. Therefore no selection bias is present, but we do agree that was not clearly stated in the original version of the manuscript. We have therefore made the following modification of the manuscript: “Group 3 consisted of three consecutive patients with different eyelid tumors, who were planned to be treated surgically. These patients received an MRI protocol that had been optimized for evaluation of eyelid tumors. When available, the CT images were included in the evaluation (Table 3)”.

Reviewer 2 Report
Some suggestions / text to be clarified:
23: in terms of identification of all eyelid layers: or structures?
24: histology: biopsy?
29: Pathology: please use another term
52: please provide more indicative detail about T-staging (i.e. dimensions / i.e. American Joint Committee on Cancer Staging (AJCC)) for the most common malignancies and the correlation between it and clinical staging. Are there any official therapeutic guidelines or major studies in relation to TNM?
53: based solely on physical examination: Please check bibliography and add references. What about biopsy?
54 examination: add (.)
56 However, these evaluations can only be accurately performed with imaging.: Evidence / Bibliography? Please point out here that the tarsus and the orbital septum are not clinically accessible.
69 In Group 1 normal eyelid anatomy was assessed on MRI by evaluating MR-images: In Group 1 normal eyelid anatomy was assessed by evaluating MR-images
70 As the eyelids of these patients were not affected by the UM: is this proved by physical examination? Eyelids metastases from uveal melanoma is an exceptional finding, however some cases are reported in the literature.
78 planned slices: available slices may be more appropriate
80 was assessed on CT in 19 patients who received volumetric CT scans: please rephrase the whole sentence. Have you conducted volumetric analysis?
81 with at least one non-affected orbit: non-affected by what?
92 were evaluated on MRI and on CT: were evaluated using both MRI and CT
114 as described earlier [14]: only the reference was mentioned. Τhis phrase can be skipped.
122-131 please check tenses
144 until: to
145 until: to
146: was evaluated on: was evaluated using
147: The bone evaluation on patient #1 from Group 3 was evaluated: please rephrase
157: images were compared with the normal eyelid anatomy on histology: the meaning is not clear. Please be more specific. With the corresponding histologic images?
164: compared to histology: of the biopsy of the lesions? Method of biopsy?
170: all eyelid layers could be identified: which are the common imaging features of tarsal plate and orbital septum - how can they be identified/defined easily?
181: Normal eyelid anatomy on the sagittal plane on MR T1-WI (A) and T2-WI (B), on CT: please check prepositions/rephrase
182: on pathology (D): please rephrase and add details
185: Further notice the uveal melanoma, located inferiorly, at E and F: where exactly on the image (arrow)?
219: please provide more imaging features depending on the sequence.
222: inferior rectus: inferior rectus muscle
227: The final pathology: the final histopathological examination
236-249: please move this text to the image caption
251: Both showed an enhancing lesion: please provide more imaging features depending on the sequence.
255: On MRI no invasion of the medial wall of the orbit was seen.: please rephrase.
256: the definitive histology: the final histopathological examination
258: growth: extension
262: pathology: histopathological examination
263-278: please move this text to the image caption
279: showed post-surgical changes at the medial inferior eyelid (not shown) due to resection: please rephrase (not shown)
285: Pathology: Histopathology
289: (dedicated eyelid protocol): should be deleted or moved to 291.
291-300: please move this text to the image caption
304: arising in: arising from
305: for the carcinoma, the malignant melanoma and the Merkel cell carcinoma. Which carcinoma? BCC? SCC?
313: with imaging playing an important role for that matter.: Evidence/bibliography? - Or rephrase
317: these important clinical indications on image: please rephrase. Are there any official clinical indications for imaging?
320-342: please move this text to introduction, seeing that it relates to Figure 1.
343: has scarcely been subject of: has scarcely been a subject of
349: reconstructions with 1 mm: please rephrase
350-359: please move this text to results, seeing that it provides information about one of the main objects of this study, the imaging features of the normal eyelid
375-377: please add this information to results, seeing that it illustrates imaging features of the normal eyelid
390-393: please add this information to results, seeing that it illustrates imaging features of the normal eyelid
400: Literature on eyelid tumors will mention the role of imaging: check tenses
421: MRI and CT images in the characterization of eyelid tumors: MRI and CT images in the delineation of eyelid tumors
424: inferior rectus: inferior rectus muscle
426: was missed: was not evident
441: not visible on the scans used to plan the higher resolution acquisitions: please rephrase
452: imaging and pathology: tomographic imaging and histopathological examination
456: identification of most of the eyelid structures is possible: how about their boundaries?
460: will: could
Author Response
Response to Reviewer 2 Comments:
We thank the Reviewer for the evaluation of the manuscript entitled “MR and CT Imaging of the Normal Eyelid and its application in Eyelid Tumors” we submitted to Cancers. You will find that the comments and critics have been taken in utmost consideration. At the “Responses” the “Lines” we mention are the Lines in the Revised Version of the Manuscript (CLEAN Version – without trackchanges). We have addressed them in the following way:
Point 1:Moderate English changes required.
Response 1:It was done.
Point 2: Introduction must be improved. Research design, Methods and Conclusions can be improved.
Response 2: Introduction, Materials and Methods and Conclusions were improved according to both reviewers’ suggestions.
Point 3: Line 23: in terms of identification of all eyelid layers: or structures?
Response 3: Line 23: The authors agree with the reviewerand used structures instead of layers.
Point 4:Line 24: histology: biopsy?
Response 4:Line 24: Tumor extension on image was compared with the results of histopathology after surgery and not with the results from a biopsy. But the abstract had been changed and this sentence had been removed.
Point 5:Line 29: Pathology: please use another term.
Response 5:Lines 27 and 29: Histopathology is now used instead of Pathology in these two lines but also through the rest of the manuscript.
Point 6:Line 52: please provide more indicative detail about T-staging (i.e. dimensions / i.e. American Joint Committee on Cancer Staging (AJCC)) for the most common malignancies and the correlation between it and clinical staging. Are there any official therapeutic guidelines or major studies in relation to TNM?
Response 6:This paragraph has been changed. Details about the T-staging are found in the Discussion.
There are therapeutic guidelines based on the TNM staging for the basal cell carcinoma and for the squamous cell carcinoma, but both not specific for the eyelid. There is also correlation between TNM and prognosis – for instance references 32 and 34.
Point 7:Line 53: based solely on physical examination: Please check bibliography and add references. What about biopsy?
Response 7: This sentence was removed.
Point 8:Line 54: examination: add (.)
Response 8:This paragraph has been changed due to suggestions from Reviewer 1, and this specific sentence was removed.
Point 9: Line 56: However, these evaluations can only be accurately performed with imaging.: Evidence / Bibliography? Please point out here that the tarsus and the orbital septum are not clinically accessible.
Response 9:This paragraph has been changed to: Lines 76-79: ”The T-staging of eyelid tumors encompasses determination on whether there is invasion of eyelid structures, such as the tarsal plate and orbital septum, which are not clinically accessible and therefore imaging can be crucial for an accurate evaluation”.
Point 10:Line 69: In Group 1 normal eyelid anatomy was assessed on MRI by evaluating MR-images: In Group 1 normal eyelid anatomy was assessed by evaluating MR-images.
Response 10:Line 93: We did not change the sentence because we want to emphasize that it is the normal eyelid anatomy on MRI. Similarly in Group 2 we assessed the normal eyelid anatomy on CT.
Point 11: Line 70: As the eyelids of these patients were not affected by the UM: is this proved by physical examination? Eyelids metastases from uveal melanoma is an exceptional finding, however some cases are reported in the literature.
Response 11:There was no evidence of melanoma in the ipsilateral superior or inferior eyelids, neither on physical examination nor on MRI.
Point 12: Line 78: planned slices: available slices may be more appropriate.
Response 12:We took this in consideration and changed it in Line 102. Similarly we also changed it in Line 114.
Point 13: Line80: was assessed on CT in 19 patients who received volumetric CT scans: please rephrase the whole sentence. Have you conducted volumetric analysis?
Response 13:Line 105: We took this in consideration and removed the word “volumetric”. It refers to CT scans with a volumetric acquisition. That is explained later, in Line 168.
Point 14: Line 81: with at least one non-affected orbit: non-affected by what?
Response 14:Line 106: We meant non-affected by disease. As it can be confusing we changed “non-affected” to “healthy” orbit.
Point 15: Line 92: were evaluated on MRI and on CT: were evaluated using both MRI and CT.
Response 15:Lines 117-9. This paragraph has been changed due to suggestions of Reviewer 1: “Group 3 consisted of three consecutive patients with different eyelid tumors, who were planned to be treated surgically. These patients received a MRI protocol that had been optimised for evaluation of eyelid tumors.When available, the CT images were included in the evaluation”.
Point 16: Line 114: as described earlier [14]: only the reference was mentioned. Τhis phrase can be skipped.
Response 16:Line 141: We took this in consideration and removed the sentence “as described earlier”.
Point 17: Line 122-131: please check tenses.
Response 17:Lines 147-160. The tenses were checked. The reason the description of the protocol sequences is in the present tense is because it corresponds to a protocol we developed but that we continue to use.
Point 18: Line 144: until: to.
Response 18:Line 172: It was changed.
Point 19: Line 145: until: to.
Response 19:Line 173: It was changed.
Point 20: Line 146: was evaluated on: was evaluated using.
Response 20:Line 174: It was changed.
Point 21: Line 147: The bone evaluation on patient #1 from Group 3 was evaluated: please rephrase.
Response 21:Line 175-6: It changed the sentence to “On patient #1 from Group 3 the evaluation of the bones was performed using 0.5 mm bone reconstructions both on the axial and coronal planes”.
Point 22: Line 157: images were compared with the normal eyelid anatomy on histology: the meaning is not clear. Please be more specific. With the corresponding histologic images?
Response 22:Line 184: We agree with the reviewer and changed the sentence to “These MR and CT images were compared with histologic slices of normal eyelids”.
Point 23: Line 164: compared to histology: of the biopsy of the lesions? Method of biopsy?
Response 23:Line 191: We changed the sentence to “compared with histopathologic examination after surgery”. The lesions were not biopsed but surgically resected.
Point 24: Line 170: all eyelid layers could be identified: which are the common imaging features of tarsal plate and orbital septum - how can they be identified/defined easily?
Response 24:That is extensively described in the Results: lines 205-11 and shown on Figures 2, 3 and 4.
Point 25: Line 181: Normal eyelid anatomy on the sagittal plane on MR T1-WI (A) and T2-WI (B), on CT: please check prepositions/rephrase
Response 25:Line 224: We checked the sentence, the prepositions seem correct but we added “,” to be more clear.
Point 26: Line 182: on pathology (D): please rephrase and add details
Response 26:Line 225: We changed pathology for histopathological examination. On Figure 2D the details are shown by arrows pointing to the normal anatomical structures of the eyelid.
Point 27: Line 185: Further notice the uveal melanoma, located inferiorly, at E and F: where exactly on the image (arrow)?
Response 27:Asterisks were added as suggested.
Point 28: Line 219: please provide more imaging features depending on the sequence.
Response 28:Line 272: In all three patients what is crucial to evaluate on MRI is the extension of the tumor within the eyelid. The characteristics of the tumor in terms of signal intensity on the various MRI sequences are in this study less important. We describe the tumor as being heterogeneous and enhancing, which should be sufficient for the T-staging.
Point 29: Line222: inferior rectus: inferior rectus muscle.
Response 29:Line 275: It was corrected.
Point 30: Line227: The final pathology: the final histopathological examination.
Response 30:Line 280: It was corrected.
Point 31: Line236-249: please move this text to the image caption.
Response 31:Lines 290-303: That is indeed supposed to be part of the Figure Caption.
Point 32: Line251: Both showed an enhancing lesion: please provide more imaging features depending on the sequence.
Response 32:Line 305: See response 28.
Point 33: Line255: On MRI no invasion of the medial wall of the orbit was seen.: please rephrase.
Response 33:Line 309: We took this in consideration and rephrased it to: “ The medial wall of the orbit was intact on MRI”.
Point 34: Line256: the definitive histology: the final histopathological examination.
Response 34:Line 310: It was changed.
Point 35: Line258: growth: extension.
Response 35:Line 312: It was changed.
Point 36: Line262: pathology: histopathological examination.
Response 36:Line 325: It was changed.
Point 37: Line263-278: please move this text to the image caption.
Response 37:Lines 317-332: That is indeed supposed to be part of the Figure Caption.
Point 38: Line279: showed post-surgical changes at the medial inferior eyelid (not shown) due to resection: please rephrase (not shown).
Response 38:Line 333-5: It was rephrased to: “showed post-surgical changes at the medial inferior eyelid (not shown), due to resection of a melanotic melanoma of the bulbar conjunctiva of the medial inferior eyelid, performed four weeks earlier”.
Point 39: Line285: Pathology: Histopathology.
Response 39:Line 339: It was changed.
Point 40: Line289: (dedicated eyelid protocol): should be deleted or moved to 291.
Response 40:Line 345: We think that “dedicated eyelid protocol“ should stay, to be consistent with the Figure Captions of patients 1 and 2 – Figures 5 and 6 – where in the beginning of the legend we also mention the type of MRI protocol used.
Point 41: Line291-300: please move this text to the image caption.
Response 41:Lines 346-55: That is indeed supposed to be part of the Figure Caption.
Point 42: Line304: arising in: arising from
Response 42:Lines 359: This paragraph was changed and this word was erased.
Point 43: Line305: for the carcinoma, the malignant melanoma and the Merkel cell carcinoma. Which carcinoma? BCC? SCC?
Response 43:Lines 361-5: The paragraph was changed to: ”Eyelid carcinomas including the basal cell carcinoma, squamous cell carcinoma, sebaceous carcinoma and other rare carcinomas such as all varieties of sweat gland carcinoma have a T-staging. Eyelid melanomas, depending on whether they arise within the skin or the conjunctiva, are staged according to the classification for skin melanoma or conjunctival melanoma respectively. Merkel cell carcinoma is staged using the Merkel cell carcinoma staging system”.
Point 44: Line313: with imaging playing an important role for that matter.: Evidence/bibliography? - Or rephrase.
Response 44:Lines 368-70: The sentence was changed to; “Both dimensions and evaluation of nearby structures’ invasion are not always possible through physical examination alone, with imaging needed for an accurate assessment. The following reference was added: Amin, M.B. et al The Eighth Edition AJCC Cancer Staging Manual: Continuing to build a bridge from a population-based to a more “personalized” approach to cancer staging. CA Cancer J Clin 2017, 67, 93-9930.
Point 45: Line317: these important clinical indications on image: please rephrase. Are there any official clinical indications for imaging?
Response 45:Lines: 370-378: It was rephrased. The T staging of the different eyelid tumors is according to UICC based solely on physical examination or after excision, with no mention to imaging. The AJCC also considers imaging in specific situations, mainly to assess invasion of the orbit, of the periorbital structures and perineural spread, but imaging evaluation of the tarsal plates and orbital septa is never specifically mentioned.
Point 46: Line320-342: please move this text to introduction, seeing that it relates to Figure 1.
Response 46:Lines 37-65: It was done.
Point 47: Line343: has scarcely been subject of: has scarcely been a subject of.
Response 47:Line 385: It was changed.
Point 48: Line349: reconstructions with 1 mm: please rephrase.
Response 48:Lines 391-2: It was rephrased to “On CT, a slice thickness reconstruction of 1 mm is suitable for the evaluation of the eyelids”.
Point 49: Line350-359: please move this text to results, seeing that it provides information about one of the main objects of this study, the imaging features of the normal eyelid.
Response 49:Lines 199-214: It was done.
Point 50: Line375-377: please add this information to results, seeing that it illustrates imaging features of the normal eyelid.
Response 50:Lines 205-9: It was done.
Point 51: Line390-393: please add this information to results, seeing that it illustrates imaging features of the normal eyelid.
Response 51:Lines 209-11: It was done.
Point 52: Line400: Literature on eyelid tumors will mention the role of imaging: check tenses.
Response 52:Line 430: It was done.
Point 53: Line421: MRI and CT images in the characterization of eyelid tumors: MRI and CT images in the delineation of eyelid tumors.
Response 53:Line 451: It was changed.
Point 54: Line424: inferior rectus: inferior rectus muscle.
Response 54:Line 455: It was changed.
Point 55: Line426: was missed: was not evident.
Response 55:Line 456: It was changed.
Point 56: Line441: not visible on the scans used to plan the higher resolution acquisitions: please rephrase.
Response 56:Line 473: It was rephrased to “not visible on the low resolution images that are used to plan the higher resolution acquisitions”.
Point 57: Line452: imaging and pathology: tomographic imaging and histopathological examination.
Response 257:Line 483: It was changed.
Point 58: Line456: identification of most of the eyelid structures is possible: how about their boundaries?
Response 58:Line 487-8: It was changed to “delineation”.
Point 59: Line460: will: could.
Response 59:Line 491: It was changed to: “may have”.

Round 2
Reviewer 1 Report
The authors have done a good job responding to the reviewer comments. I think it is fair to publish this work now.